# Strategies for Producing Low FODMAPs Foodstuffs: Challenges and Perspectives

**DOI:** 10.3390/foods12040856

**Published:** 2023-02-17

**Authors:** Fernanda Galgano, Maria Cristina Mele, Roberta Tolve, Nicola Condelli, Maria Di Cairano, Gianluca Ianiro, Isabella D’Antuono, Fabio Favati

**Affiliations:** 1School of Agricultural, Forestry, Food and Environmental Sciences (SAFE), University of Basilicata, 85100 Potenza, Italy; 2Clinical Nutrition Unit, Fondazione Policlinico Universitario A. Gemelli IRCCS, 00168 Rome, Italy; 3Department of Translational Medicine and Surgery, Università Cattolica del Sacro Cuore, 00168 Rome, Italy; 4Department of Biotechnology, University of Verona, 37134 Verona, Italy; 5Digestive Disease Center, Fondazione Policlinico Universitario A. Gemelli IRCCS, 00168 Rome, Italy; 6National Council of Research, Institute of Science of Foods Production (CNR-ISPA), 70126 Bari, Italy

**Keywords:** irritable bowel syndrome, fermentable carbohydrates, functional food, dough fermentation, bioactive compounds

## Abstract

In recent years, there has been a growing interest in a diet low in fermentable oligosaccharides, disaccharides, monosaccharides, and polyols (FODMAPs) as a promising therapeutic approach to reduce the symptoms associated with irritable bowel syndrome (IBS). Hence, the development of low FODMAPs products is an important challenge for the food industry, and among the various foodstuffs associated with the intake of FODMAPs, cereal-based products represent an issue. In fact, even if their content in FODMAPs is limited, their large use in diet can be an important factor in developing IBS symptoms. Several useful approaches have been developed to reduce the FODMAPs content in processed food products. Accurate ingredient selection, the use of enzymes or selected yeasts, and the use of fermentation steps carried out by specific lactic bacteria associated with the use of sourdough represent the technical approaches that have been investigated, alone or in combination, to reduce the FODMAPs content in cereal-based products. This review aims to give an overview of the technological and biotechnological strategies applicable to the formulation of low-FODMAPs products, specifically formulated for consumers affected by IBS. In particular, bread has been the foodstuff mainly investigated throughout the years, but information on other raw or processed products has also been reported. Furthermore, taking into account the required holistic approach for IBS symptoms management, in this review, the use of bioactive compounds that have a positive impact on reducing IBS symptoms as added ingredients in low-FODMAPs products is also discussed.

## 1. Introduction

Irritable bowel syndrome (IBS) is a chronic gastrointestinal disease identified by recurrent abdominal pain associated with defecation together with a change in bowel habits [1]. IBS is very common and has a worldwide prevalence of 9% in males and 14% in females [2]. Based on the Rome IV criteria, which classifies IBS in four subtypes depending on the predominance of the stool form, the most common (40–60%) subtype is the one with predominance of diarrhea (IBS-D) [3,4]. The etiology of IBS is still unknown, but it appears to be multifactorial and includes different pathogenic pathways involving impairment of the brain–gut axis, visceral hypersensitivity, alteration of the immune system, derangement of gut microbiome, and psychosocial factors. Because of the high incidence of anxiety and depression in functional disorders [5], IBS is associated with decreased quality of life [6,7] and has a notable impact on healthcare systems [8,9,10,11]. The pharmacological therapeutic options for IBS are still unsatisfactory [12]; therefore, the interest of physicians and patients for dietary interventions is rising in last years. Some foods are known to be poorly tolerated by patients with IBS [13,14], and modifications in the intake of fibers and other triggers (i.e., caffeine, alcoholic beverages, and fats) represent a common therapeutic advice for IBS [15,16,17,18,19]. Furthermore, a gluten-free diet appears to provide an amelioration of symptoms in patients with diarrhea-predominant IBS [20,21,22], although with alternate results [23]. More recently, the restriction in the intake of FODMAPs has appeared as a promising therapeutic option in patients with IBS, as supported by an increased body of evidence [24]. In Europe, owing to the lack of legislation, very few products with a low FODMAPs labelling are available on the market and, as mentioned before, IBS patients use gluten-free products, generally low in FODMAPs.

However, for IBS patients, gluten elimination is not strictly necessary and not even recommended considering the lacking sensory appeal, structure, and nutritional value [25]. Thus, the development of sensorially accepted low FODMAPs breads and cereal- based foods, produced through a careful selection of the raw material as well as considering the fiber inclusion and exploiting the FODMAPs reduction via a bioprocessing approach, is an emerging area of research.

Within this context, this review aims to explore the approaches that have been used to produce low FODMAPs cereal-based products so far. Furthermore, the use of bioactive molecules able to mitigate IBS symptoms is discussed in the view of future strategies to consider low FODMAPs food development.

## 2. FODMAP Classifications

The acronym FODMAPs was coined by Gibson and Shepherd [26] and linked to a low FODMAPs diet that has been shown to improve the IBS pathology, as reported by extensive clinical research.

The FODMAPs have common characteristics at the level of metabolic fate (Table 1). In fact, they are poorly absorbed in the small intestine, and then arrive unbroken in the colon, where are rapidly metabolized by the bacterial microflora [27]. During colonic fermentation, there is gas production (hydrogen and methane), which leads to an extension of the colon, and a consequent increase of water in the lumen that, finally, can lead to diarrhea.

*FOS and GOS*. The FOS and GOS are the main FODMAPs ingested through diet, and they are widely present in cereals and legumes (wheat, spelt, barley, rice, chickpeas, lentils, peas) and in some vegetables (garlic, onion, cabbage) [28]. They are large molecules, composed by oligosaccharides with different degrees of polymerization, which are also based on their natural or enzymatic source. In functional foods manufacturing, FOS and GOS are added for their prebiotic properties but also to exert beneficial actions such as reducing constipation and controlling body weight or as low-calorie sweeteners [29] (Figure 1).

*Lactose*. Lactose is the main disaccharide present in mammalian milk and its derivatives; it is formed by galactose and glucose and linked by β(1-4) glycosidic bond. Generally, lactose is digested in the intestine by a lactase enzyme present on the intestinal brush border, releasing two monosaccharides. In patients with lactase deficiency, the lactose reaches the colon unbroken where is metabolized by intestinal microflora and can be classified as a FODMAP [30]. Yang et al. [31] observed that intestinal gas production was proportional to the amount of lactose ingested, and these symptoms increased in patients with IBS. However, to date, the effects of lactase deficiency on symptoms of irritable bowel syndrome (IBS) are still not well defined [32] (Figure 1).

*Fructose.* The dietary sources of fructose are mainly fruits and honey. In the intestinal lumen, fructose is present as free hexose or following hydrolysis of sucrose and it can included as a FODMAP when it is not equimolar with glucose. This is because of its two-absorption mechanism; by the GLUT5 on the brush border membrane, however this is a low capacity transport, and by GLUT2, which is responsible for carrying both fructose and glucose out of the enterocyte across the basolateral membrane [33]. The excess of fructose drives to its malabsorption and colonic metabolism which, through fermentation, can lead to osmotic diarrhea, gas, and bloating [34] (Figure 2).

*Polyols.* Polyols are sugar alcohols naturally present in some fruits, vegetables, mushrooms, and sugar-free sweeteners; they are formed by catalytic hydrogenation of carbohydrates and in several products, such as chewing gum, candies, and beverages, are used as alternatives to sucrose, yielding fewer calories per gram, low blood glucose response, and protecting against tooth decay [35]. Polyol human metabolism goes through a passive diffusion and absorption (about 33%) in the small intestine, with an absorption rate depending on both the individual patient and the molecular size of the polyol [36]. To date, it is still unclear if polyol ingestion increases the symptoms of the disease in patients with IBS [37]. However, the slow absorption of sorbitol and mannitol leads to their rapid metabolism by the colonic microflora with a consequent increase of luminal water content and production of gas, triggering abdominal symptoms [38] (Figure 2).

## 3. FODMAPs Content in Foods

The management of IBS should take into account the challenge of balance the high increase rates of IBS and the possibility of introducing new foods fully fitting into the low FODMAPs diet, which is an eating practice that allows for improving this pathology [39]. FODMAPs are found in a wide variety of foods, such as breads, grains, nuts, legumes, fruits, vegetables, and sweets. An accurate estimate of the FODMAPs content in a food is difficult to do, as a lot also depends on how the foods are cooked, processed, and administered at the same time [40]. However, the FODMAPs content levels has been provided by several scientific investigations that deal with this issue and actually, it is possible to differentiate between foods with a high and low content of FODMAPs (Table 2) [41].

Ispiryan et al. [25] quantified and characterized FODMAPs in cereals and legumes as wheat substitutes in dietary foods. In particular, legumes, such as peas and broad beans, contained high amounts of GOS (from 4.48 to 4.87 g/100 g dry matter), whereas fructans were absent. Instead, wheat and other cereals (rice, barley) were rich in FOS (from 1.38 to 3.61 g/100 g dry matter). The same authors proved that some pseudo-cereals such as buckwheat were FODMAPs-free, however, contained other non-digestible soluble sugars (fagopyritols) not included in the FODMAPs classes, but their similar implication in IBS syndrome cannot be excluded. In the study reported by Liljebo et al. [42], a database was implemented by adding to the nutritional data FODMAPs’ most common food items, in order to make easier the estimation of daily FODMAPs intake, which took place over 4 days using 117 Swedish individuals.

In the beverages, fruits, and vegetables most commonly consumed, fructose was present in grams, whereas the major fructan intake arose from the wheat, rye, fruit, and vegetable consumption. The lactose was derived from the intake of milk and yogurt. The GOS came from bread, both wheat and rye and, finally, the intake of polyols was also derived from fruits. As result of the study, the average daily FODMAPs intake of 19 g was calculated, and the data is in agreement with other studies performed on healthy populations. Although some limitations have to be considered in this study, such as seasonal variation, stage of ripeness, and genetic variation of the analyzed samples, the investigation allowed for recommending guidelines on monitoring the dietary intake of FODMAPs and to have a list of the most common foods, to facilitate their inclusion or exclusion into a Low FODMAPs diet [42,43]. Updating needs and acquiring database-organized data on FODMAPs content is a challenge that concerns in particular the functional foods sector.

## 4. Cereal Product Formulations with Low FODMAPs Content for Consumers with IBS

For the targeted development of a low-FODMAPs product, there are specific cut-off levels related to the content of oligosaccharides (fructans and galacto-oligosaccharides), sugar polyols (mannitol and sorbitol), lactose, and fructose in excess of glucose, which should be considered and used as a benchmark [40]. These cut-offs have been validated and established based on the findings of clinical studies, considering the typical serving size of food consumed as a singular meal that triggered the symptoms in IBS patients. For grains, legume, nuts, and seeds oligosaccharides (fructan and α-galacto-oligosaccharides), the cut-off value is <0.30 g per serving. For vegetables, fruits, and all other products, the oligosaccharides cut-off is <0.20 g per serving and the same cut-off is true for sorbitol or mannitol, whereas for total polyols it is <0.40 g per serving. When fructose in excess to glucose is the only FODMAP present (for fruit or vegetables), the cut-off is <0.40 g per serving instead when there are other FODMAPs, when the value for fructose in excess to glucose is <0.15 g per serving. Finally, for lactose, the cut-off level is <1.00 g per serving size.

### 4.1. Approaches to Produce Low FODMAPs Cereal-Based Products

Although cereals and cereal-based products have a modest content of FODMAPs, their regular consumption makes them the primary sources of FODMAPs intake in the western diet [27,44,45]. Cereal-based foods are products that most affect IBS symptoms. Almost half of the IBS patients declared that bread contributes to the symptomatology [18]. In many cultures, bread is a staple food, and its high daily consumption represents a significant proportion of FODMAPs intake [46].

Consequently, among the different cereal-based foods, the primary interest is the formulation of low-FODMAPs bread. As a matter of fact, to date, little or nothing has been reported on the development and characterization of low FODMAPs cereal-based products other than bread. Some authors characterized commercial low FODMAPs cereal-based products, including biscuits, crackers, and breadsticks, but have not posted about their development [25]. The only exception is represented by the research recently conducted by Radoš et al. [47] to develop sensory attractive high-fibre and low-FODMAPs crackers using different gluten-free wholemeal cereals, oilseeds, plant protein, sweet potato, sourdough, and spices, and by Habuš et al. [48], who produced cereal-based 3D printed snacks. The FODMAPs content in the cereal-based product can be obtained by enforcing different strategies.

One approach is the avoidance of high FODMAPs ingredients with the selection of appropriate grain species, variety, and level of refinement [49]. Another way to reach a low FODMAPs product is the degradation of these compounds achieved through pre-treatments on raw grains or flours. Otherwise, different biotechnological approaches can degrade the FODMAPs during the production process, such as the use of sourdough culture, the enzymatical removal of fructo- and galacto-oligosaccharides, and the choice of yeast species used for dough fermentation [50]. These approaches can be used alone or in combination. Table 3 summarizes the state of the art relating to the effect of the use of different approaches for the reduction of FODMAPs content in cereal and legumes flours, breads, and other cereal- and legume-based products.

### 4.2. Ingredients Selection

In food ingredients, there are two main classes of FODMAPs: fructans in gluten-containing cereals (wheat, spelt, barley, rye) and α-galactooligosaccharides (GOS) in pulses (peas, lentils, chickpeas, etc.) [25,68]. Wheat flour contains about 1.53% of fructans and higher levels can be found in bran because they are generally stored in the outer grain layers [45]. In rye, fructan levels can even reach 6.6% [50]. Although the amount of fructans in spelt, which is a primitive wheat, has been reported to be about 1.29% [69], many individuals stated that foods produced with spelt flour were easily digested compared to the wheat counterparty [49] to such an extent that some authors suggested consuming spelt products instead of bread wheat [27]. Biesiekierski et al. [44] suggested that, albeit slightly, the lower FODMAPs content in spelt products compared to wheat might plausibly be the reason for these observations. To clarify to which extent the different environments could affect the FODMAPs content of the final wheat product, Ziegler et al., quantified it in flours obtained by different variety of grain species grown in different locations [49]. Specifically, bread wheat (*Triticum aestivum* L.), spelt (*T. spelta* L.), durum (*T. durum* DESF.), emmer (*T. dicoccum* SCHRANK), and einkorn (*T. monococcum* L.) were analyzed. The authors found out that FODMAPs levels ranged from 1.2 to 2.1% in emmer and einkorn flours, respectively and, unexpectedly, did not observe differences in the FODMAPs content of the bread produced with wheat and spelt species under investigation. Similarly, Longin et al., studied different wheat varieties to compare their FODMAPs content [59]. The authors evaluated the fructan levels in the flour of 21 wheat varieties. Wheat fructans content varied from 0.8 to 2.0% with a mean of 1.21%. The fructans content of emmer was comparable to wheat. By contrast, spelt had a significantly higher mean fructans content than wheat and emmer, with the spelt varieties ranging from 1.3 to 1.8%. Einkorn had the highest average amount of fructans (values ranging from 1.6 to 3.6%). Furthermore, this research highlighted that the FODMAPs content of wholegrain flours was much higher than one of bread produced with the same variety, thus, suggesting a reduction effect of the production process on these compounds. The FODMAPs average content of the investigated 21 wheat varieties was 1.21 g/100 g, whereas the obtained bread had an average lower than 0.4 g/100 g DM, with a reduction of more than 65%.

Gluten-free cereals (oat, millet, and rice), as well as the pseudo-cereals (quinoa and buckwheat), do not contain any of the FODMAPs commonly investigated. However, buckwheat accumulates other soluble carbohydrates (fagopyritols) that may act as FODMAPs. Fagopyritols are non-digestible, fermentable, and structurally similar to GOS and may have a similar effect on a sensitive gut [25]. Regarding pulses, they accumulate GOS (raffinose, stachyose, verbascose, ajugose; predominantly stachyose) at levels from 1% up to >10% [25]. Pulses, such as lentils and chickpeas, are commonly used for the development of gluten-free and health-promoting products because of their technological properties and nutritional benefits, such as their high protein content [70]. However, lentils contain high levels of GOS, ranging from 1.8 to 7.5% and, hence, contribute to the total FODMAPs intake [51]. Similarly, chickpeas have a high content of α-GOS (5–10%) with raffinose, stachyose, verbascose, and ciceritol being mainly represented [71]. Recently, Escobedo et al., assessed the α-GOS content in black and pinto beans, reporting a concentration of 3.3 and 4.8%, respectively [54]. The authors highlight the possibility to reduce the content by using a thermal and enzymatic treatment. As mentioned before, in one serving there should be no more than 1 g of lactose and 0.4 g of polyols to avoid the symptomatology in patients affected by IBS [40]. Thus, the incorporation of polyols, such as glycerol, sorbitol, and mannitol, to improve the bread texture and the shelf-life and the use of dairy ingredients in breadmaking for their nutritional benefits and functional properties, must be avoided [72,73].

The need to not use ingredients with a high FODMAPs content does not imply a ban on taking substances isolated from these ingredients, such as proteins and starches isolated from cereals and legumes. Wheat flour isolate, as an example, is used for the formulation of low FODMAPs foods [74]. Given the high nutritional quality and the good amino acidic composition, recently, Joehnke et al., proposed the use of lentil protein isolate to produce low FODMAPs products [67]. The researchers found that the ultrafiltration technique was useful for the removal of compounds with a molecular weight lower than 10 kDa reducing, in this way, the GOS concentration by more than 90%, from ~4 g to 0.37 g/100 g of dry matter. Instead, the acid extraction coupled with isoelectric precipitation was able to diminish the levels of FODMAPs in fava beans [75]. When the upstream selection of ingredients with low FODMAPs content is carried out, it is not necessary to apply any biotechnological treatments for their reduction.

### 4.3. Enzymatic FODMAPs Reduction

The main FODMAPs of cereals and pulses can be degraded by enzymatic activity. Thus, to obtain the decrease of FODMAPs in foods, it is possible to add enzymes, exploiting the microbial enzymatic activity during fermentation or activate endogenous seed enzymes [76]. Fructans degradation in the raw materials can be achieved through inulinases that catalyze the hydrolysis of β(2–1) glycosidic linkages of inulin, exoinulinases, and endoinulinases specific for the terminal linkages of inulin and the internal β(2–1) glycosidic linkages, respectively, and also the invertase has been shown to catalyze inulin degradation by an exo-mechanism. Some authors have reported the possibility of using the α-galactosidase treatment of pulses flour to cleave the indigestible α-galactosyl linkages in GOS (raffinose, stachyose, verbascose) [77]. In a recent study, Nyyssölä et al., reported the potential of a new α-galactosidase for galactooligosaccharides hydrolysis in peas and fava bean-based prototypes (meat analogues, crackers, spoonable products) to lower GOS levels by up to 90% [52]. Also commercially available is an enzyme-based low-FODMAPs solution, the patented LOFO™ enzyme, which claims to decrease the fructans content of grain products by more than 50% [78]. With a view to the reduction of FODMAP content of food ingredients, Atzler et al., applied enzymatic treatments with ß-fructofuranosidases, inulinase, and α-galactosidase to extracts obtained from whole meal wheat and lentil flours reporting a high potential for enzymes treatments to reduce FODMAPs content [51]. Specifically, inulinase degraded GOS and fructans with over 90% degradation, whereas α-galactosidase fully decomposed GOS.

However, it should be considered that this study was done on extracts and not on food where the interaction and the presence of other components could change the level of degradation. In addition, treatment with ß-fructofuranosidases and inulinase leads to the production of melibiose, manninotriose, and manninotetraose, which possess characteristics of FODMAPs and might also cause IBS symptoms. Thus, it is necessary to link the enzymatic oligosaccharide hydrolysis with a strategy for reducing the resulting degradation products. In view of reducing the FODMAPs content of cereal-based products, malting can be applied as a delivery system of endogenous enzymes [53]. Different seeds were malted and their FODMAPs content was investigated [53]. Results showed a significant reduction, up to 80–90%, of GOS in lentils and chickpeas, whereas buckwheat did not contain any of the FODMAPs usually considered. Concerning cereals, namely barley and wheat, it was suggested to employ a combined approach of malting and fermentation since malting only is not enough to significantly reduce the FODMAPs content.

### 4.4. Reduction Mediated by Yeast and LAB Fermentation

During dough fermentation, fructans are hydrolyzed by the enzymes expressed by yeast and, according to the strain, through the dosage and the fermentation time, the fructans’ hydrolysis may amount to 80–90% [50]. Saccharomyces cerevisiae, the most used leavening agent in industrial baking, can express invertase and it has been reported to significantly reduce the fructans in wheat flour up to 90%. Ziegler et al., using a commercial baker’s yeast, reported a decrease of up to 90% in FODMAPs, highlighting the importance of the fermentation time rather than the variety of wheat used [49]. After 1 h of fermentation, the authors reached a fructan degradation of 60%, but with ∼1% excess of fructose in bread. When increasing the fermentation time to 4.5 h, fructan degradation reached 90% with only 0.03% excess fructose. This clarification is because there is also a limit to the so-called free fructose (i.e., the fructose over glucose), which must be less than 0.5 g per 100 g of food [76]. The efficiency of the fermentation process depends on the amount of fructans in the matrix. For example, Li et al., demonstrated the limitation of yeast fermentation to achieve FODMAPs reduction in rye-based bread compared to wheat-based bread, owing to the higher fructan concentration in the rye [66]. S. cerevisiae was also used in co-culture with *Kluyveromyces marxianus* to obtain the fructan reduction in wheat-based bread [56]. The cited authors reported that *K. marxianus* was more active in wheat fructans degradation than *S. cerevisiae*; however, they opted for the co-culture to improve the bread loaf volume, which is otherwise penalized because of the low amounts of CO_2_ produced by *K. marxianus* yeast. Additionally, the use of sourdough was proven to be an effective method to reduce the content of fructans in bread. Compared to yeast-based fermentation, sourdough fermentation exploits the action of both lactic acid bacteria (LAB, generally Lactobacillus strains) and yeast (mainly *Saccharomyces cerevisiae*, *Candida humilis*, *Wickerhamomyces anomalus*, *Torulaspora delbrueckii*) present at a ratio of 10:1 to 1000:1 [58]. Since yeast and certain LAB utilize the saccharides enzymatic degradation products, the synergistic effect of added enzymes, yeast, and LAB is a strategy to efficiently reduce the total FODMAPs content as recently demonstrated [65,66]. It is possible that the longer leavening time required for the sourdough compared to the baker’s yeast results in more extensive hydrolysis of the cereal carbohydrates, such as FODMAPs, during sourdough fermentation. However, because of the high LAB variety and behavior under different environmental conditions, the application of sourdough technology is more complex compared to yeast fermentation [79]. Pejcz et al., reported a decrease in fructan content with increasing fermentation times both in bread produced with whole and light wheat flour [60]. The best performances were obtained with the use of sourdough and the addition of a pure culture of *Lactobacillus plantarum* (now *Lactiplantibacillus plantarum*) and 150 min fermentation time. In contrast, Pejcz et al., did not find a reduction of fructan content in rye bread, and the extension of fermentation time did not reduce fructan content in bread [61]. Anyway, their results confirm the effectiveness of the use of sourdough for the reduction of fructans. In addition, Menezes et al., indicated that it is possible to apply sourdough fermentation for producing low FODMAPs bakery products [46]. Indeed, they reported a fructans reduction from 69 to 75% in sourdough-fermented bread compared with yeast-fermented. Schmidt and Sciurba also evaluated the effect of proofing time and sourdough use on bread [64]. The authors reported a reduction of fructans and, consequently, of FODMAPs content attributed to fermentation time. At the same time, they found that the sourdough did not decrease the total FODMAPs content but changed the FODMAPs profile, with a decrease in fructans and an increase in mannitol content. According to the authors, the production of low FODMAPs rye bread requires special attention and represents a big challenge because of the need to employ different processing technologies. Recently, Shewry et al., (2022) investigated the effect of yeast and sourdough-based fermentation on the FODMAPs content of three types of wheat [62]. The results indicated that yeast fermentation led to a greater reduction in FODMAPs content compared to sourdough fermentation, despite what was previously reported.

Most of the research dealing with the investigation of methods to reduce FODMAPs content in cereal-based products refers to flour pretreatment and bread formulations. Moreover, some research is aimed at the reduction in other products, mainly crackers and 3D-printed snacks [47,52]. Indeed, Habuš et al., applied a combined approach of inulinased yeast fermentation on wheat and amaranth bran to reduce their fructans content and use them in the production of 3D printed snacks [48]. They obtained a fructans reduction of up to 93% depending on the bran type and bioprocessing agent. These recent investigations show the rising interest of the research community in developing low-FODMAPs foods other than bread and in the years to come, this could indicate greater attention also from food manufacturers in the production of foods suitable to help physicians and IBS patients in their therapeutic strategy.

## 5. New Approaches in Formulating Cereal-Based Products for IBS Management

Dietary fiber, phytochemicals, and vitamins have been proven to reduce IBS symptoms [80,81]. To date, different evidence showed the effectiveness of these compounds, mostly used as supplements, in the management of IBS symptoms. These results open the possibility to studies aimed at the formulation of foods where these compounds or their sources are incorporated into cereal-based, and non-, functional foods intended for IBS patients.

From a nutritional point of view, the exclusion of FODMAPs causes inevitable disadvantages. A more difficult intake of dietary fiber and the possible compromise of gut health correlated to the lowered diversity of the gut microbiota is one of the main drawbacks. This is because of the shared characteristics between dietary fiber and FODMAPs, such as fermentability and hygroscopic behavior. Frequently used soluble, non-viscous fermentable or un-fermentable dietary fiber, such as GOS and FOS, as well as inulin, are often harmful to IBS patients. In addition, also insoluble, fermentable dietary fiber is not suitable for a low FODMAPs diet. Therefore, only a few types of dietary fiber can be used as alternatives. Soluble fiber, such as psyllium and calcium polycarbophil are indicated as effective in IBS, but bran generally seems not to be beneficial [82].

Recently, Balmus et al. [80] highlighted that the potential employers of many forms of fruit, vegetable, and medicinal plant waste could be an important source of bioactive molecules that can be helpful in IBS management Indeed, the effect on IBS symptoms of different nutraceuticals has been evaluated by different authors. Peppermint oil showed promising results thanks to its antispasmodic properties [83,84]. Other clinical studies evaluated the effect of compounds such as aloe vera, ginger, red pepper, polydatin, curcumin, and tumeric as reviewed by Fifi et al., and Chen et al. [83,85]. The mitigation of IBS symptoms has been obtained and nevertheless, in some cases, the differences were not significant with the control and the studies present different limitations. Furthermore, the use of single bioactive phyto-compounds such as phlorizine showed beneficial action against IBS syndrome, acting on the suppression of the pro-inflammatory cytokines present in rats affected by IBS [86]. A large class of phyto-compounds and polyphenols and their metabolites have long been known for their anti-inflammatory, antioxidant, anticancer, and immunomodulatory properties and are to be considered as promising bioactives in the fight against different gastrointestinal diseases. Moreover, polyphenols can positively modulate the bacterial population of the intestinal microbiota, thus improving the host’s immune system, and consequently the general health of the intestine [87]. Among vitamins, vitamin D supplementation improved IBS patient symptomatology, with reports that vitamin D_3_ and isoflavone from soy seed administrated to IBS patients could reduce inflammation and gut permeability [80,81,82,83,84,85,86,87,88,89,90].

Therefore, further investigations are needed to confirm the potential use of these compounds in an IBS diet and their eventual addition to the production of functional foods.

## 6. Conclusions

In IBS patients, the control and management of the gastrointestinal symptoms is of the utmost importance to improving their quality of life. Often a low-FODMAPs diet is recommended, reducing or eliminating specific fermentable carbohydrates, such as oligosaccharides, disaccharides, monosaccharides, and polyols. However, although food labels report basic data about the fiber and carbohydrate content, the information provided does not allow IBS patients to properly select suitable foods, given the fact that in the fiber and carbohydrates groups different moieties can be found, eventually affecting the IBS patients’ health. Nowadays, only a few food items are labelled as being low-FODMAPs products and in several countries, a specific legislation is lacking.

In order to overcome the problem, IBS patients often follow a gluten-free diet, despite not being celiac, just because of the wider availability of these products. However, gluten elimination is not strictly necessary and, considering the low nutritional, sensory and rheological characteristics of these foodstuffs, it is not even recommended. Thus, the availability of tailored food items that have a low-FODMAPs content is highly needed, while maintaining desirable nutritional, sensory, and hedonic characteristics.

Ingredient selection, enzymatic approach, reduction mediated by yeast, LAB fermentation, and the use of sourdough, applied alone or in combinations, have been proven to be effective methods to reduce the FODMAPs content in cereal-based products. A further step could be represented by the use as ingredients in low-FODMAPs products of compounds able to mitigate IBS symptoms, such as fibers (e.g., ispaghula), polyphenols, vitamin D, and peppermint oil. However, although these compounds have been proven to be effective, no studies can be found in the literature dealing with their incorporation in low-FODMAPs food items.

In conclusion, much more research is needed in order to develop and adequately formulate low-FODMAPs products, taking into account both the health and satisfaction issues of the IBS patients. A close cooperation between food scientists dealing with food product development issues and medical researchers involved in clinical trials should be sought, so as to define useful technologies and specific limits that may then allow the food industry to invest in producing the required products.

## Figures and Tables

**Figure 1 foods-12-00856-f001:**
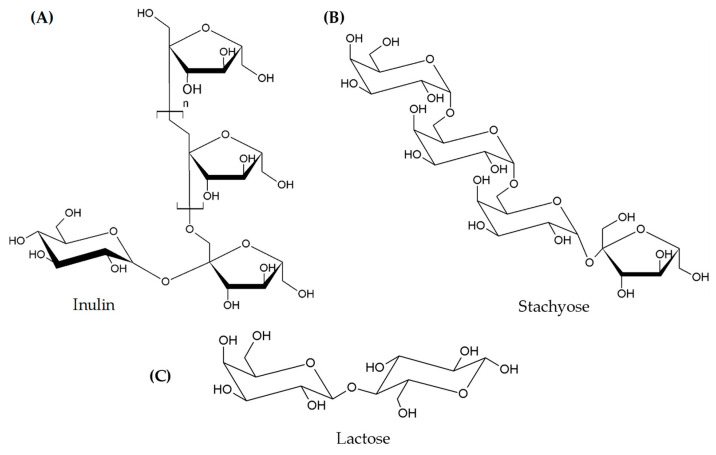
Examples of FODMAPs’ chemical structures. (**A**) Fructans: e.g., inulin composed by fructose units linked β(2-1) glycosidic bonds, with a total polymerization degree according to the origin plant; (**B**) Galacto-oligosaccharides: e.g., stachyose consisting of sucrose that has an α-D-galactosyl-(1→6)-α-D-galactosyl moiety attached at the 6-position of the glucose; (**C**) disaccharides: e.g., lactose made up of one unit each of glucose and galactose, joined by β-1, 4 glycosidic linkage.

**Figure 2 foods-12-00856-f002:**
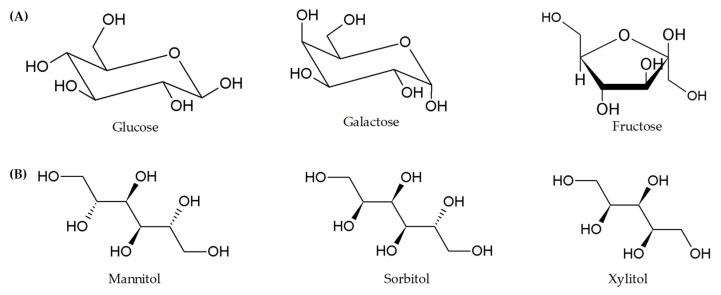
Examples of FODMAPs’ chemical structures. (**A**) Monosaccharides; (**B**) polyols.

**Table 1 foods-12-00856-t001:** Classification of the fermentable oligo-, di-, mono-saccharides, and polyols (FODMAPs).

Classes	Examples
Oligosaccharides	Fructans (FOS), Galacto-oligosaccharides (GOS)
Polysaccharides	Lactose
Monosaccharides	Fructose
Polyols	Sorbitol, mannitol, xylitol, erythritol, polydextrose, and maltitol

**Table 2 foods-12-00856-t002:** Examples of low and high FODMAP foods, based on the standard serving size.

Food	Low FODMAPs	High FODMAPs
Fruits	Kiwifruit, blueberry, banana, mandarin, orange, passionfruit, grapefruit.	Peaches, apples, pears, watermelon, cherries, mango, apricots.
Vegetables	Carrot, celery, lettuce, eggplant, zucchini, green beans, bok choy.	Asparagus, Brussels sprout, cabbage, fennel, mushrooms, onion, garlic.
Dairy	Brie/camembert cheese, feta cheese, lactose-free milk.	Cow, sheep and goat milk, ice cream, yoghurt, ricotta, cottage.
Grain/cereals	Gluten-free bread/cereal products, sourdough spelt bread, quinoa/rice/corn pasta.	Pasta, wheat bread, biscuits, couscous.
Sweeteners	Maple, rice malt and golden syrups, sucrose.	Honey, high fructose corn syrup, sorbitol, mannitol, xylitol.

**Table 3 foods-12-00856-t003:** Effect of the use of different approaches for the reduction of FODMAPs content in cereal and legumes flours, bread, and other cereal- and legume-based products.

Approaches	Product	Type of Flour	Results	Reference
Ingredient selection and fermentation time	Bread	Wheat	Prolonged proofing time (>4 h) ↓ FODMAPs content up to 90%	Ziegler et al. [49]
Ingredients selection	Cracker	Wholemeal, buckwheat, millet, and white maize	High fibre, low-FODMAP product	Radoš et al. [47]
Enzymatic (β-fructofuranosidases and α-galactosidases)	Flour	Wholemeal wheat and lentil flours (degradation on FODMAPs extract)	Inulinase degraded over 90% GOS and fructans α—galoctosidase degrade 100% GOS invertase low degradation yield	Atzler et al. [51]
Enzymatic (α-GOS)	High moisture meat analogues Crackers Spoonable product	Faba bean and yellow been	↓ GOS over 90%	Nyyssölä et al. [52]
Enzymatic (activation of endogenous enzymes by malting)	Grains	Spring malting barley, wheat, chickpeas, oat, lentils, buckwheat	Malting ↓ 80–90%GOS in lentils and chickpeas—fructans not synthetized in oat barley and wheat malts slightly higher fructans content	Ispyrian et al. [53]
Enzymatic (α-galactosidases) and soaking treatmentand thermal treatment	Flour	Common bean	Ezymatic hydrolysis (α-GOS) and soaking and thermal treatment ↓ GOS up to 97.6%	Escobendo et al. [54]
Yeast fermentation (inulinase producer)	Bread	Wheat flour	*Kluyveromyces marxiaus* strain ↓ 90% fructans level; *Saccharomices cerevies* ↓ 56% reduction; co-culture of the two-strain leads to a bread low FODMAP and good loaf volume	Struyf et al. [55]
Yeast fermentation (30 *K. marxianus* strains)	Bread	Wheat	*Kluyveromyces marxianus* strain CBS6014 can degrade more than 90% of the fructans	Struyf et al. [56]
Yeast fermentation (28 *S. cerevisiae* strains)	Bread	Wholewheat	Final fructan level of 0.3% dm, Strains with a low invertase activity yielded fructan levels around 0.6% dm. The non-bakery strains produced lower levels of CO_2_ in the bread	Laurent et al. [57]
Yeast fermentation	Model system	Different rye and sourdough as yeast source	*Saccharomyces cerevisiae* isolated from Austrian traditional sourdough showed the highest degree of degradation of the total fructan content and the highest gas building capacity, followed by *Torulaspora delbrueckii*	Fraberger et al. [58]
Yeast fermentation and fermentation time	Bread	Wheat (21 varieties)	Different wheat varieties differ up to 5 times in their potential to form FODMAPs in bread. FODAMPs content tend to be lower in long fermentation but not significant FODMAPs reduction >65%	Longin et al. [59]
Yeast fermentation and enzymatic (inulinase)	3D printed snack	Wheat and amaranth bran	↓ fructan content up to 93%	Habuš et al. [48]
Yeast and LAB (sourdough) fermentation and ingredients selection	Bread	Light and whole wheat	Sourdough and extended fermentation time ↓ fructans content use of light flour ↓ fructans content	Pejcz et al. [60]
Yeast and LAB fermentation (sourdough) and ingredients selection and fermentation time	Bread	Rye flour (endosperm and whole meal	Sourdough ↓ fructans content— prolonged fermentation time no effect on fructans content	Pejcz et al. [61]
Yeast and LAB (sourdough) fermentation	Bread	Wheat, rye, emmer	Wheat bread ↑ fibre and fructan contents compared to other flours- Yeast fermentation ↑ reduction of fructans and raffinose	Shewry et al. [62]
Yeast and LAB (sourdough) fermentation	Bread	Wheat flour	Sourdough ↓ FODMAPs-sourdough bread not best tolerated by IBS patients than yest fermented	Laatikainen et al. [63]
LAB (sourdough) and fermentation time	Bread	Wheat flour and rye flour (wholemeal and refined)	Prolonged proofing time ↓ fructans content sourdough changed FODMAPs composition by ↓ fructans content and ↑ mannitol content-refined wheat flour bread meets low FODMAPs criteria—rye and whole meal wheat flour high FODMAPs regardless of processing condition employed	Schmidt and Sciurba [64]
LAB fermentation (25 fructophilic lactic acid bacteria strains)	Bread/dough	Durum wheat	Fermenting dough resulted in lower loaf volumes	Acín Albiac et al. [65]
LAB (sourdough) fermentation	Bread	Wheat	Sourdough ↓ of fructans up to 69–75%	Menezes et al. [46]
LAB (sourdough) fermentation + *Lactobacillus crispatus*	Bread	Rye and wheat flour	Sourdough fermentation with *L. crispatus* ↓ fructans more than 90% Conventional sourdough fermentation ↓ fructans (65–70%)	Li et al. [66]
Other approaches isoelectric precipitation and ultrafiltration	Lentil protein isolate	Lentil	↓ GOS 58% isoelectric precipitation ↓ GOS 91% ultrafiltration	Joehnke et al. [67]

↑: increase; ↓: reduction; dm: dry matter.

## Data Availability

Data is contained within the article.

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
