# Peer review of "Strategies for Producing Low FODMAPs Foodstuffs: Challenges and Perspectives"

_foods, 2023, doi:10.3390/foods12040856_

Round 1
Reviewer 1 Report
Dear Editor and Authors,
The topic is interesting and innovative. I agree with the authors' opinion that so far there has been little or nothing reported on the development and characterization of low FODMAP cereal-based products other than bread. The order of the chapters is well planned. There is only a need for minor changes which are described below.
ABSTRACT
Please rewrite the abstract for too many introductory sentences. It will be better if there is one introductory sentence at the beginning, then the purpose and scope of the work, results and conclusions.
INTRODUCTION
The introduction is well written, only the end lacks a demonstration of the novelty of the subject matter. Please show how many review articles on the low-Fodmap diet or products (see references examples) have been published so far. What is new in this article compared to the previous ones?
Conclusions
Conclusions are well written but too long and contain sentences that do not provide specific information, e.g. The sentence "In the last 15 years, several useful approaches have been developed to reduce the FODMAPs content in food products, espe-397 cially in bread in which , despite the modest FODMAPs content, its regular consumption 398 makes it the primary sources of FODMAPs." can be deleted or shortened and combined with the next sentence.
References
Literature sources are very well selected, but it is still worth reviewing the proposed review articles and adding them to the list.
Bellini, M., Tonarelli, S., Nagy, A. G., Pancetti, A., Costa, F., Ricchiuti, A., … Rossi, A. (2020, January 1). Low FODMAP diet: Evidence, doubts, and hopes. Nutrients. MDPI AG. https://doi.org/10.3390/nu12010148
Vandeputte, D., & Joossens, M. (2020, November 1). Effects of low and high FODMAP diets on human gastrointestinal microbiota composition in adults with intestinal diseases: A systematic review. Microorganisms. MDPI AG. https://doi.org/10.3390/microorganisms8111638
Zannini, E., & Arendt, E. K. (2018). Low FODMAPs and gluten-free foods for irritable bowel syndrome treatment: Lights and shadows. Food Research International, 110, 33–41. https://doi.org/10.1016/j.foodres.2017.04.001
Author Response
ABSTRACT
Q: Please rewrite the abstract for too many introductory sentences. It will be better if there is one introductory sentence at the beginning, then the purpose and scope of the work, results and conclusions.
A: The Authors thank the Reviewer for the suggestion, and the abstract has been revised. However, the Authors feel that the suggested organization (introduction, purpose of the work, results and conclusions) would be out of the scope of an abstract for a review paper.
INTRODUCTION
Q: The introduction is well written, only the end lacks a demonstration of the novelty of the subject matter. Please show how many review articles on the low-Fodmap diet or products (see references examples) have been published so far. What is new in this article compared to the previous ones?
A: The Authors thank the Reviewer for the suggestions, and they believe that although the topic linking FODMAPs to IBS pathology is a widely discussed issue, in this review they wanted to report an overview of the papers that have dealt technological and biotechnological approaches to reduce the FODMAP content in cereal products. They also included the paragraph on bioactive compounds as an approach for food sector to reduce the content of FODMAPs in particular for functional foods. The interesting reviews suggested by the Reviewer address the problem more from a "medical" point of view, with a briefly discussion on the aspect of food development.
Conclusions
Q: Conclusions are well written but too long and contain sentences that do not provide specific information, e.g. The sentence "In the last 15 years, several useful approaches have been developed to reduce the FODMAPs content in food products, especially in bread in which , despite the modest FODMAPs content, its regular consumption makes it the primary sources of FODMAPs." can be deleted or shortened and combined with the next sentence.
A: The Authors thank the Reviewer for the suggestions. However, taking into account also the comments made by the Reviewer #2, the whole section has been rewritten as requested.
Reviewer 2 Report
This review paper was well-written, but I still have some comments for this manuscript.
1. I suggested the author increase the section or paragraph of introduction.
2.I suggested the author increase some data into the table and contex of manuscript.
3.The conclusion should be re-ritten.
4.The aim or important of this manuscript was not clear.
Author Response
Q: I suggested the author increase the section or paragraph of introduction.
A: The introduction has been edited by briefly adding the aim of the review. Anyway, the introduction has not been increased since it is brief on purpose; it just served as a premise to introduce the context in which the development of low FODMAPs foodstuff is places.
Q: I suggested the author increase some data into the table and contex of manuscript.
A: The Authors thank the Reviewer and have made changes to the manuscript in various sections, thanks to the suggestions received. With regard to the tables, they do not consider to further "burden" some of them, for example for table 2 many numerical data (cut-offs of the various categories of FODMAPs) have been already reported in the text (paragraph 4).
Q: The conclusion should be re-ritten.
A: Following the Reviewer suggestion this section has been rewritten.
Q: The aim or important of this manuscript was not clear.
A: The Authors thank the Reviewer and the aim has now been highlight in the abstract and introduction.
Reviewer 3 Report
This review paper can be useful and impressive if the authors manage to provide specific information. For example, the title is not emphasize as in the abstract stated the review is focused on bread. Maybe the authors can delete the word 'bread' in the abstract section. More comments are attached in the file.

Author Response
Comments and Suggestions for Authors
This review paper can be useful and impressive if the authors manage to provide specific information. For example, the title is not emphasize as in the abstract stated the review is focused on bread. Maybe the authors can delete the word 'bread' in the abstract section. More comments are attached in the file.
|
Title |
The title of the manuscript should be concise. The abstract stated that this manuscript is specific focused on bread. |
|
|
The whole manuscript is discussed on cereal-based products instead of bread. |
|
Line 344 |
The sub-topic is not suitable for this manuscript as it is discussed on bioactive compound, not specifically used for bread production. |
A: The Authors thank the Reviewer. The title and abstract have been rewritten according to the useful suggestions. About the subtopic dealing with the possible inclusion of bioactive compounds, although to date are ingredients not used in the formulation of low-FODMAPs products, the Authors feel that it represents a future perspective and new approaches for the formulation of low-FODMAPs products.
Reviewer 4 Report
Review on manuscript (foods-2144089):
"Production strategies for the low FODMAPs cereal-based food to manage irritable bowel syndrome”
by Fernanda Galgano, Maria Cristina Mele, Roberta Tolve, Nicola Condelli, Maria Di Cairano, Gianluca Ianiro, Isabella D’Antuono*, Fabio Favati
submitted to Foods
This review aims to shed light on the current state of knowledge of the technological and biotechnological strategies applicable to the formulation of low-FODMAPs products suitable for IBS suffers, with a specific focus on bread. In addition, also the inclusion of valuable bioactive compounds useful for IBS management as low-FODMAPs food ingredients is discussed. Generally, the manuscript is readable, but the manuscript needs some corrections
Detailed recommendation:
Abstract: please add more data to the abstract.
Key words: add: irritable bowel syndrome
Add more information about chemical structure of FODMAPs
Line 63: table x ???
Author Response
Detailed recommendation:
Q: Abstract: please add more data to the abstract.
A: The Authors thank the Reviewer for the suggestion, and the abstract has been revised.
Q: Key words: add: irritable bowel syndrome
A: The Authors thank the Reviewer and the key word “irritable bowel syndrome” has been added.
Q: Add more information about chemical structure of FODMAPs
A: The Authors following the Reviewer's suggestion have added two figures to better illustrate as example the chemical structures of some FODMAP compounds
Q: Line 63: table x ???
A: The Authors thank the Reviewer and the mistake has been corrected.